# Physical Properties of Fast-Growing Wood-Polymer Nano Composite Synthesized through TiO_2_ Nanoparticle Impregnation

**DOI:** 10.3390/polym14204463

**Published:** 2022-10-21

**Authors:** Istie Rahayu, Wayan Darmawan, Deded Sarip Nawawi, Esti Prihatini, Rohmat Ismail, Gilang Dwi Laksono

**Affiliations:** 1Department of Forest Products, Faculty of Forestry and Environment, IPB University, Kampus IPB Dramaga, Bogor 16680, Indonesia; 2Department of Chemistry, Faculty of Mathematics and Natural Sciences, IPB University, Kampus IPB Dramaga, Bogor 16680, Indonesia

**Keywords:** impregnation, mangium, TiO_2_ nanoparticle, photocatalyst, physical properties

## Abstract

Mangium (*Acacia mangium* Willd.) is a fast-growing wood that is widely grown in Indonesia. The impregnation method is needed to improve the qualities of the wood. In this study, TiO_2_ nanoparticle (79.17 nm) was produced using the hydrothermal method. The purpose of this study was to analyze the effect of TiO_2_ nanoparticle impregnation on the density and dimensional stability of mangium and the effectiveness of the presence of TiO_2_ nanoparticle in wood in degrading pollutants. The mangium samples (2 cm × 2 cm × 2 cm) were placed inside impregnation tube. The impregnation solutions included water (untreated), 1% TiO_2_ nanoparticle, and 5% TiO_2_ nanoparticles. The samples were analyzed for density, weight percent gain (WPG) dan bulking effect (BE). Samples were also analyzed by X-ray diffraction (XRD) and Fourier-transform infrared spectroscopy (FTIR). TiO_2_ nanoparticle resulted in an increase in density, WPG, and BE-treated mangium. Based on XRD and FTIR results, TiO_2_ nanoparticle was successfully impregnated into mangium wood. Scanning electron microscopy–energy-dispersive X-ray spectroscopy analysis indicated that TiO_2_ nanoparticle covered the surface of the wood cells. The TiO_2_-impregnated mangium wood has a higher photocatalyst activity than untreated, indicating better protection from UV radiation and pollutants.

## 1. Introduction

The need for quality wood that is strong, and durable remains very high in Indonesia and elsewhere. Wood is irreplaceable raw material for both inside (interior) and outside (exterior) applications. Based on data [1], sawn wood production in Indonesia reached 2.6 million m^3^ in 2020. The demand for industrial wood under the Indonesian Forest Concession Association (APHI) [2] reached 2 million m^3^ in 2019. Mangium (*Acacia mangium* Willd.) wood is widely used for making door and window frames and as raw material for furniture in Indonesia. Mangium wood has a specific gravity value of 0.61 (0.43–0.66) and belongs to the strong class II–III [3]. This fast-growing wood has several disadvantages that hinder its use, such as high-water content as well as hygroscopic properties that enable it to absorb water easily. The water content in wood can affect dimensional stability and mechanical properties [4]. According to Chu et al. [5], reducing the hygroscopic properties of wood is important because it can improve low-quality wood by increasing its dimensional stability. Several methods can be used to improve wood quality, including wood modification, drying, and preservation.

In this study, we use a wood impregnation method. This treatment inserts chemicals into the pores, lumens, and cell walls of wood, where they precipitate and stored without damaging the wood. In addition, the impregnation process involves chemical reaction between the impregnation compound and the most reactive groups in the wood cell wall [6,7]. Many studies have been conducted on impregnation of fast-growing wood to improve the quality, using chemicals such as methyl methacrylate [8,9,10], phenol formaldehyde [11,12], monoethylene glycol [13], melamine formaldehyde and furfuryl alcohol [14], and nano Fe_3_O_4_ [15].

The impregnation method using TiO_2_ nanoparticle (NP) can also increase the wood’s resistance to damage caused by weathering and UV radiation. In particular, the use of TiO_2_ nanoparticle offers an alternative formulation for better protection, and Fuva and Hovde [16] reported that surface modification using TiO_2_ nanoparticle can increase the wood’s UV resistance. The resistance is because nanoparticle have a higher surface area. Wang et al. [17] showed that TiO_2_ nanoparticle anatase is more suitable as a photocatalyst, while rutile TiO_2_ performs better in blocking UV irradiation. TiO_2_ serves as a highly functional materials, and it has been widely used in paints [18] because of its superior chemical stability and non-toxicity [18,19]. TiO_2_ nanoparticle has also been used in wood to increase dimensional stability [20], fire resistance [21], rot protection [22], and prevent weathering [23,24,25,26,27].

In some previous studies, TiO_2_ nanoparticle was mostly purchased from commercial sources. However, in this study we synthesized TiO_2_ nanoparticle via hydrothermal method. This hydrothermal method involves conducting the reaction with water as the solvent in a closed system at a certain temperature and pressure [28]. Liu et al. [29] highlighted that following the reaction in a closed container, the contents can be recovered and reused after cooling to room temperature, which makes this method environmentally friendly. Because the hydrothermal method with water solvent enables controlling the morphology, thermal stability, and crystalline phase, it has been often used to synthesize nano-sized materials including TiO_2_ photocatalyst [30,31]. The hydrothermal process has several advantages: such as the use of simple equipment, catalyst-free growth, low cost, large uniform production area, low process temperature, and ease of controlling particle size. Further, particle properties such as morphology and size can be controlled by adjusting the reaction temperature and time and concentration of precursors [32]. Based on the results of previous studies, information was obtained that TiO_2_ nanoparticle have an increase in reactivity as photocatalysts when forming composites with other compounds such as toluene degradation reactions as organic volatile pollutants with Co_3_O_4_/TiO_2_ nanocomposite [33,34], hydrocarbon compounds with RuO_2_/TiO_2_ nanocomposite [35], and ethanol with Pt/TiO_2_ nanocomposite [36]. The purpose of this study is to analyze the effect of impregnation of TiO_2_ nanoparticle on the density and dimensional stability of mangium and to investigate the effectiveness of the TiO_2_ nanoparticle presence in wood as a polymer nanocomposite for degrading pollutants.

## 2. Materials and Methods

Ten-year-old mangium wood samples were obtained from community forests in the Bogor area, West Java, Indonesia. The mangium wood is then cut into 100 cm long pieces. All samples came from the same tree and were cut at the same time to obtain uniform wood samples. Chemicals used in this study included bulk TiO_2_ anatase (Pure^®^), ethanol (Mallinckrodt chemical^®^), and deionized water.

### 2.1. Sample Preparation

Mangium wood was cut using a chain saw and table saw without distinguishing between sapwood and heartwood. The test samples each measured 2 cm × 2 cm × 2 cm [37], and consisted of 30 samples, with 10 replications at each level. The sample were used in testing the weight polymer gain (WPG), leachability (L), anti-swelling efficiency (ASE), water uptake (WU), bulking effect (BE), and density.

### 2.2. Synthesis of TiO_2_ Anatase Nanoparticle Using the Hydrothermal Method

The preparation of TiO_2_ nanoparticle solution was carried out using the hydrothermal method. Seven grams of pure TiO_2_ powder was placed in a beaker, and then mixed with 75 mL of distilled water. The solution was then reacted in a Teflon stainless steel autoclave at 75 °C for 4 h, after which it was gradually cooled to room temperature over 24 h. The resulting precipitate was separated onto a porcelain dish and then washed with ethanol and calcined for 3 h at 500 °C [38].

### 2.3. Impregnation Method

The wood samples were oven dried before the impregnation process and then weighed and their dimensions were measured. Furthermore, the impregnation process is carried out using an impregnation tube. Wood samples were put into each container and an impregnation solution was poured which consisted of untreated, 1% (*w*/*v*) TiO_2_ nanoparticle, and 5% (*w*/*v*) TiO_2_ nanoparticle. The samples were under vacuum (−0.7 bar) for 30 min and then under pressure (1 bar) for 2 h. The test sample was removed from the tube and dried in the oven at 65 °C for 12 h and then left in the oven at 103 °C until the weight was constant. The impregnated samples were then used in density and dimensional stability test and the effectiveness of the presence of TiO_2_ nanoparticle in wood in degrading pollutants was analyzed.

### 2.4. Physical Properties

Samples were weighed and measured to obtain their weight and volume both before and after the impregnation process. The weights and measurements were used to calculate the following values: weight percent gain (WPG), bulking effect (BE), density (ρ), anti-swelling efficiency (ASE), water uptake (WU), and leachability (L)

The WPG was calculated using Equation (1): WPG (%) = [(W_1_ − W_0_)/W_0_] × 100(1)
where W_0_ is the initial oven-dried weight of the sample before impregnation, and W_1_ is the oven-dried weight of the sample after impregnation.

ASE testing was carried out by repeated water soaking [4]. ASE was calculated with Equation (2):ASE (%) = [(S_u_ − S_t_)/S_u_] × 100(2)
where S_u_ is the volume shrinkage of untreated wood sample, and S_t_ is the volume shrinkage of the treated wood.

WU testing on samples was carried out after immersion in water for 24 h. WU was calculated using Equation (3):WU (%) = [(W_2_ − W_1_)/W_1_] × 100(3)
where W_2_ is the sample weight after immersion in water for 24 h.

The BE test was calculated using Equation (4):BE (%) = (V_1_ − V_0_)/V_0_ × 100(4)
where V_0_ is the initial oven-dried volume of a wood sample before impregnation and V_1_ is the oven-dried volume of wood sample after impregnation.

Oven-dried density (ρ) was calculated using Equation (5): (5)$$\rho \;(g/{\mathrm{cm}}^{3})=\frac{B}{V}\times 100$$
where B is the weight of the sample before or after impregnation treatment, and V is the volume of the sample before or after impregnation treatment.

### 2.5. Photocatalyst Activity Test

Total of 0.1 g of the test sample was put into 100 mL of 10 ppm methylene blue solution and then stirred until homogeneous. The mixed solution was irradiated under UV light for 60 min, and every 10 min, the solutions were taken to measure the absorbance at a wavelength of 666 nm by UV-VIS Spectrophotometer [39].

### 2.6. Testing the Effect of UV-Vis Radiation

The test sample was irradiated for 6 h continuously with UV-Vis radiation in a closed room. After that, the effect of radiation on wood was analyzed by FTIR [24].

### 2.7. Characterization of Synthesized TiO_2_ Nanoparticle

The synthesized TiO_2_ nanoparticle were characterized using UV-Vis spectrophotometer, FTIR, and XRD [40]. Data processing is performed with Origin 8.5 software (Northampton, MA, USA) for spectrum and diffractogram data. Inorganic molecular elucidation was carried out with QualX software (Roma, Italy) and Mercury software (Cambridge, UK).

Characterization of optical properties using UV Vis spectrophotometer (Shimadzu 1800) was carried out by dissolving TiO_2_ nanoparticle anatase at a concentration of 0.1 M in demineralized water and then measured its absorption in the UV wavelength range [41]. The results of this measurement are also used for band gap energy analysis using the Tauc method.

Characterization using FTIR (Perkin-Elmer Spectrum One) and XRD (PANAnalytical Empyrean) was carried out by filtering TiO_2_ nanoparticle anatase in powder form using a 100 mesh sieve followed by FTIR analysis of pellet method with KBr and measured at wave numbers 400–4000 cm^−1^ and XRD analysis with powder measurement method at an angle range of 2θ which is 0–90°

### 2.8. Characterization of Impregnated Mangium Wood

#### 2.8.1. Scanning Electron Microscopy and Energy-Dispersive X-Ray Spectroscopy

The penetration ability and distribution of TiO_2_ nanoparticle into wood cell walls were analyzed using SEM (ZEISS EVO’50). Untreated and impregnated wood samples were cut to a size of 0.5 cm × 0.5 cm × 0.5 cm in a tangential plane, placed on a conductor adhesive, coated with gold, and observed under SEM at a voltage of 15 kV. Wood samples were also analyzed using energy dispersive X-ray analysis (EDX) to analyze the chemical content of the wood.

#### 2.8.2. Fourier Transform Infrared Spectrometry 

Untreated and impregnated samples were ground to a particle size of 200 mesh and embedded in potassium bromide (KBr) pellets in a ratio of 1:100. The pellets were analyzed by FT-IR (Perkin-Elmer Spectrum One) with a scanning range of 4000 to 400 cm^−1^ at a resolution of 4 cm^−1^ for 32 scans.

#### 2.8.3. X-ray Diffraction Analysis

Wood samples were slashed using a 2 mm thick cutter in a tangential plane. The degree of crystallinity of the wood sample (incision) was analyzed by XRD-PANAnalytical Empyrean type with a 1D PIXcel detector. The parameters used in the device are: Cu Kα radiation with a graphite monochromator, a voltage of 40 Kv, a current of 30 mA, and a scan range of 2Ɵ between 0 and 90° with a scanning speed of 2°/min.

### 2.9. Data Analysis

This study used a completely randomized design and data were evaluated using analysis of variance (ANOVA), followed by Duncan’s test at a significance level of α = 5%. Statistical testing was performed using IBM SPSS Statistics (Statistical Package for Service Solutions) version 25.0 program Stanford, California, CA, USA.

## 3. Results

### 3.1. Synthesized TiO_2_ Nanoparticle

TiO_2_ powder forms a coarse spherical morphology with granules measuring approximately less than 100 nm. The magnification of the diffraction peak was confirmed by the small size of TiO_2_ nanoparticle (see details in XRD discussion). These nanoparticle form agglomerations with approximately 500 nm (Figure 1), which Zanatta [42] previously attributed to van der Waals forces. The agglomeration of nanoparticle originates from the induced dipoles in the synthesis method. The short-range interactions arising from induced dipoles can be easily broken by the stresses that occur during impregnation.

#### 3.1.1. FTIR Result

The analysis of the FTIR spectrum of TiO_2_ nanoparticle (Figure 2) led to the identification of the functional groups of Ti-O at a wave number of 545 cm^−1^ and Ti-O-Ti at a wave number of 802 cm^−1^, which are bonds formed in the framework of TiO_2_ compounds. The TiO_2_ spectrum showed intense peaks at 3655 and 3860 cm^−1^ due to the stretching of the OH group of the H_2_O molecule bonded to the TiO_2_ compound [43]. This finding indicates that the TiO_2_ compound has been successfully synthesized by the hydrothermal method.

#### 3.1.2. XRD Result

The XRD analysis were intended to determine the phase, degree of crystallinity, and crystal size of TiO_2_ synthesized by hydrothermal method. The diffractogram of the TiO_2_ nanoparticle sample (Figure 3) shows peaks at the following 2θ values: 25.26, 36.92, 37.76, 38.63, 48.00, 53.86, 55.03, 62.66, 68.74, 70.46, 75.03, 82.65. The calculation results show that the synthesized TiO_2_ nanoparticle have a high degree of crystallinity (99.86%). The crystal size of the TiO_2_ nanoparticle is determined from the diffractogram data using the Scherrer equation, where *λ* = X-ray wavelength used; *θ* = diffraction angle; and K = is a constant whose magnitude depends on the crystal form factor, diffraction field (hkl), and *β* is the full width at half maximum (FWHM) or integral breadth of the peak [44].

The crystal size of TiO_2_ nanoparticle calculated by the Scherrer equation (Equation (6)) is 79.17 nm. This proves that the synthesized TiO_2_ is a nanoparticle because it has a particle size in the range of 1 to 100 nm based on crystal size [45].
(6)$$D=\frac{K\lambda }{\beta cos\theta }$$

The UV spectrum analysis of TiO_2_ nanoparticle (Figure 4) indicated wavelength absorbance value of 362 nm. In TiO_2_, the valence band comes from the 2p orbital of the oxygen atom and the conduction band comes from the 3d orbital of the titanium atom [46].

The Tauc method was used for the band gap energy analysis (Figure 5) based on the assumption that the energy affected by the absorbance coefficient (α) can be determined using Equation (7):(α·*hv*)^γ^ = *B*(*hv* − *E*g)(7)
where *h* is Planck’s constant, *v* is the frequency of the proton, *E*g is the band gap energy, and *B* is the constant. The factor depends on the natural electron transition value which is equivalent to ½ or 2 for indirect and direct transitions in the band gap energy [47]. The analysis of the energy band gap based on the synthesized TiO_2_ nanoparticle yielded a value of 3.40 eV. This value is greater than the band gap value of bulk phase anatase TiO_2_ nanoparticle which is 3.22 eV. These results confirm that when the size of the nanoparticle is smaller, the band gap will be larger [40].

The crystal structure of TiO_2_ nanoparticle was determination using QualX software developed by [48] and computerized with Mercury software. Based on the results of the analysis of the crystal structure, a tetragonal crystal form was obtained (Figure 6).

### 3.2. Physical Properties of Mangium Wood

Table 1 presents the dimensional stability and density values of untreated and treated mangium wood. The density, WPG, BE, and ASE values showed an increasing trend, while the WU values showed a decreasing trend. The leaching value increased when 1% TiO_2_ nanoparticle was used, decreased with 5% TiO_2_ nanoparticle concentration. The same trend was previously shown for the parameters of dimensional stability and density [15,49].

The highest WPG value was obtained with 5% TiO_2_ nanoparticle treatment (2.76 ± 0.72)%, which also yielded the highest density value (0.57 ± 0.03)%. In addition to the increasing WPG and density values, physical properties such as BE and ASE also had an increasing trend. The highest BE and ASE values were associated with 5% TiO_2_ nanoparticle treatment (2.10 ± 1.03% and 21.76 ± 7.97%, respectively). A different trend occurred for WU value, which was lowest with the 5% TiO_2_ nanoparticle treatment (46.66 ± 6.67)%.

### 3.3. Characteristics of Mangium Wood Impregnated with TiO_2_ Nanoparticle

#### 3.3.1. SEM-EDX Analysis

Figure 7a–f shows the differences between the untreated mangium wood and samples that were treated with TiO_2_ nanoparticle. In both samples, TiO_2_ nanoparticle appeared to cover the cell walls, with deposits also being present in the intercellular cavities (yellow color in Figure 7b,c,e,f). 

Based on the results of quantitative analysis using SEM EDX (Table 2), the TiO_2_ content in mangium wood impregnated with 5% TiO_2_ nanoparticle had a higher Ti content (4.69%) than that impregnated with 1% TiO_2_ nanoparticle concentration (0.35%).

#### 3.3.2. Photocatalyst Activity Test

The concentration of methylene blue is directly proportional to its absorption value on a visible light spectrophotometer. Figure 8a shows that the presence of TiO_2_ nanoparticle in wood significantly affected absorbance. Wood that was impregnated with TiO_2_ nanoparticle had a higher photocatalytic activity than untreated, with the C/C_o_ ratio declining as the UV radiation treatment time increased.

Figure 8b shows that the role of TiO_2_ nanoparticle can be seen significantly with the percentage value of methylene blue degradation increasing with time of UV radiation until it reaches 90.64%. The 1% TiO_2_ nanoparticle (82.49%) and 5% TiO_2_ nanoparticle (74.35%)—impregnated wood had a higher photocatalytic activity than the untreated. The end result of the photocatalytic reaction is carbon dioxide gas which can be released into nature and can be utilized by plants for the photosynthesis process [50].

#### 3.3.3. Evaluation of TiO_2_ Nanoparticle Anatase Efficiency on the Degradation of Methylene Blue

Electrical energy consumption is one of the important criteria in the photochemical process because it is related to the determination of the operating cost of the treatment. The figure-of-merit is the electrical energy per order, defined as the number of kWh of electrical energy required for reducing the concentration of a pollutant by 1 order of magnitude (i.e., 90% degradation), in 1 m^3^ of the contaminated sample [51]. The electric energy per order, *E_EO_* (kWh/m^3^/order), is commonly used, which can be estimated using an equation for a batch reactor [52]. Equation (8) was as follows:(8)$$E_{E0}=\frac{P\times t\times 1000}{V\times 60\times log\;\frac{C_{o}}{C_{f}}}$$
where *P* is the electrical power of the UV lamp (kW), *t* is irradiation time (min), *V* is the volume of the treated wastewater (L), *C_o_* is the initial concentration of the pollutant (mg L^−1^), and *C_f_* is the final concentration of the pollutant (mg L^−1^). The results of the analysis (Figure 9) showed that the highest energy consumption occurred in the untreated blue methylene with a value of 6955.58 kWh/m^3^ and the lowest occurred in the anatase TiO_2_ nanoparticle sample with a value of 1166.68 kWh/m^3^. Energy consumption in untreated wood is 3679.12 kWh/m^3^, this value is higher than treated wood with TiO_2_ nanoparticle which is 2030.65 kWh/m^3^ (1% TiO_2_ nanoparticle) and 1585.59 kWh/m^3^ (5% TiO_2_ nanoparticle).

To estimate the kinetic rate of the removal methylene blue reaction, applying the pseudo-first-order (n = 1) kinetic model using the following Equation (9) [53]:(9)$$-In\;\frac{C}{C_{0}}=K_{obs}\;t$$
where *C*_0_ is the initial concentration of the pollutant, *C* is the concentration at time *t*, and *K_obs_* and *t* are the observed rate constant and the irradiation time, respectively. According to Equation (9), *K_obs_* can be determined as the slope of the plot of $-In\;\frac{C}{C_{0}}$ versus the reaction time (Figure 10). The linearity of lines (Figure 10) demonstrates that the photocatalytic kinetics for methylene blue obeyed the pseudo-first-order kinetic model. The lines on Figure 10, had high values of the correlation coefficients (R^2^ > 0.93). This finding confirmed the validity of the pseudo-first order model [54]. The calculated values for *K_obs_* were 0.0406 L/min (anatase TiO_2_ NP), 0.0062 L/min (untreated methylene blue), 0.0334 L/min (untreated wood), 0.0262 L/min (1% TiO_2_ nanoparticle), and 0.0123 L/min (5% TiO_2_ nanoparticle).

#### 3.3.4. UV-Vis Radiation Analysis

Mangium wood that has been impregnated with TiO_2_ nanoparticle is treated with UV radiation at a wavelength of 366 nm and a power of 120 watts and visible light radiation from an incandescent lamp with a power of 150 watts for 6 h in a closed system. The radiation distance used is 15 cm. This test was conducted to determine the level of protection of TiO_2_ nanoparticle against UV-Vis radiation.

The results of the FTIR spectrum analysis (Figure 11) showed a decrease in the intensity of the C-H aromatic and C=O functional groups which were part of the functional group of lignin compounds in mangium wood that were not impregnated with TiO_2_ nanoparticle by UV-Vis radiation treatment. This indicates that there is a degradation initiation reaction in the main structure of the wood starting from the adhesive compound between wood, namely lignin. The same thing did not happen to mangium wood impregnated with TiO_2_ nanoparticle which had the same spectrum pattern as untreated wood without UV-Vis radiation treatment.

#### 3.3.5. FTIR

Analysis using FTIR (Figure 12) was carried out to identify the functional groups present in untreated and treated mangium wood. The FTIR spectrum for untreated revealed the C-H functional group from the framework of the aromatic compounds at a wave number of 593 cm^−1^, the C-O functional group at a wave number of 1055 cm^−1^, the C=C functional group at a wave number of 1631 cm^−1^, the C-H functional group at a wave number of 2914 cm^−1^, and the O-H functional group at wave number 3284 and 3887 cm^−1^ [55]. All functional groups identified in untreated mangium were also identified in treated mangium with 1% and 5% TiO_2_ nanoparticle, but in treated mangium wood, functional groups were identified from Ti-O at wave numbers 523 and 533 cm^−1^, and functional groups Ti-O-Tiat wave numbers 696 and 719 cm^−1^, indicating the bonds formed in the framework of TiO_2_ compounds [43].

#### 3.3.6. XRD

XRD (Figure 13) was used on mangium wood samples impregnated with TiO_2_ nanoparticle to identify its characteristics through phase analysis of cellulose and TiO_2_ and to determine TiO_2_ crystal size in mangium wood. The diffractogram of mangium wood impregnated with 1% TiO_2_ nanoparticle showed peaks at a value of 2θ namely 25.25, 36.62, 38.09, 39.04, 47.83, 53.93, 55.03, 62.78, 69.13, 70.42, 75.38, and 83.05. In the diffractogram of mangium wood impregnated with 5% TiO_2_ nanoparticle, there were peaks at 2θ values, namely 25.34, 36.54, 37.72, 39.09, 47.73, 53.66, 55.32, 62.72, 68.95, 70.29, 74.88, and 82.68. In the diffractogram of untreated mangium wood, peaks for 2θ occurred at 15.29, 21.975, and 34.31. In the diffractogram of mangium wood impregnated with 1% TiO_2_ nanoparticle, there were peaks at 2θ values, namely 15.77, 22.20, and 34.39. In the diffractogram of mangium impregnated with 5% TiO_2_ nanoparticle, there were peaks at 2θ values, namely 15.35, 21.97, and 34.47. 

The XRD analysis indicated that mangium wood impregnated with 1% TiO_2_ nanoparticle experienced an increase in the value of the degree of crystallinity to 74.17% while mangium wood impregnated with TiO_2_ nanoparticle 5% experienced a decrease in the degree of crystallinity to 65.13% (Table 3).

## 4. Discussion

### 4.1. Synthesized TiO_2_ Nanoparticle

#### XRD Analysis

A comparative analysis with the standard anatase TiO_2_ diffractogram JCPDS card number 78-2486 [40] confirmed anatase TiO_2_ compounds were successfully synthesized by the hydrothermal method. The degree of crystallinity of TiO_2_ nanoparticle was determined by comparing the crystalline lattice with the total number of lattices in the diffractogram (amorphous and crystalline). The degree of crystallinity is very important because it is linearly related to the reactivity and optical properties of a compound [56]. These results in Figure 4 are in accordance with the research of Li et al. [57], showing that TiO_2_ nanoparticle synthesized by the hydrothermal method have a maximum wavelength value in the UVA range of 320–400 nm. Maximum absorbance occurs when electrons are excited from the valence band to the conduction band [46].

The optical properties of a material can be determined through its interaction with electromagnetic radiation fields consisting of transmission, absorption, emission, reflection, refraction, diffraction, or scattering effects. In the special case of semiconductors such as TiO_2_, this optical property can be tested in the ultraviolet, visible, and infrared wavelength ranges. This ability is related to the characteristics of the compound’s band gap energy [58]. A smaller particle size is associated with a larger band gap, as fewer molecular orbitals are added to the possible ground state of the particle’s energy. Therefore, absorption will occur at higher energies, so a shift toward shorter wavelengths will be seen [40]. Our findings (Figure 6) are in agreement with the study of Opoku et al. [59], who reported that among three TiO_2_ phases namely rutile, anatase, and brookite, anatase was the phase that had the best photocatalytic activity with a body-centered tetragonal crystal structure, lattice parameters a = b = 3.782 and c = 9.502, and the space group I 41/amd.

### 4.2. Physical Properties of Mangium Wood

The observed increase in the WPG value indicated that TiO_2_ nanoparticle was successfully dispersed in the wood, filling its cavities and cell walls of wood caused the weight of mangium wood to increase from untreated to treated as the TiO_2_ concentration increased. Based on the report by Chu et al. [5], an increased WPG value indicates the presence of TiO_2_ nanoparticle that is deposited in the cell cavities and fills them. The WPG value increased along with the increase in the density value. The increased density value was probably caused by the TiO_2_ nanoparticle solution that was successfully dispersed into the wood, which caused the cavities and cell walls of the wood to become denser and thicker compared to untreated. As previously reported, the higher the density value of the treated wood, the more content in the wood cavities and the thicker the wood cell walls [60].

The increase in the ASE value is directly proportional to the BE value. This shows that the addition of TiO_2_ nanoparticle made the wood cell walls expand and increased bulking, resulting in improved dimensional stability with treatment. Hill [6] showed that the higher the BE value, the more polymer filled in the wood cell wall which could increase the dimensional stability of the wood. This is consistent with a report by Devi and Maji [61] showing that the ASE value increases when TiO_2_ nanoparticle was added. This result was attributed to TiO_2_ nanoparticle being deposited into the empty space of the wood and making the cell walls thicker. This also causes the density value to increase and the shrinkage and swelling to be reduced. These changes therefore improved the dimensional stability of the wood.

Chu et al. [5] reported that the increased WPG value caused the WU value to decrease when wood was impregnated with TiO_2_ nanoparticle. Based on Rathnam et al. [62], the value of WU WPC (wood polymer composite) decreased with the addition of nanoclay and TiO_2_ nanoparticle. Like nanoclay, TiO_2_ nanoparticle also provided barrier properties to prevent the wood from absorbing water. In addition, impregnated wood with 5% TiO_2_ nanoparticle had a better leaching value than that with 1% TiO_2_ nanoparticle, but between the two concentrations it shows a small possibility for leaching because the leaching value produced is quite low. Pouya and Younes [63] reported that at the end of a 6-day test period, almost no leaching of TiO_2_ nanoparticle at any concentration occurred, compared with preservatives such as zinc sulfate and boron which are difficult to form and easy to leach and remove from wood.

### 4.3. Characteristics of Mangium Wood Impregnated with TiO_2_ Nanoparticle

#### 4.3.1. SEM-EDX Analysis

According to Liu et al. [64], TiO_2_ nanoparticle are deposited evenly on the cell walls of the wood instead of accumulating in the pores. This nanoparticle interaction may be due to physical interactions or hydrogen bonds that occur between the OH present in the cell wall polymer and the hydrolyzed TiO_2_ [17,65]. Mangium wood impregnated with 5% TiO_2_ nanoparticle appeared to have more of the surface of the wood cells covered. This is in accordance with the physical properties test that the increase in WPG was directly proportional to the increase in the concentration of TiO_2_ nanoparticle impregnated into mangium wood.

#### 4.3.2. Photocatalyst Activity Test

Pollutants can increase the degradation rate of biopolymer materials [66]. Pollutants consist of two types, namely organic and inorganic pollutants [67]. The use of titanium oxide compounds can protect biopolymer materials such as wood from UV radiation due to the strong ability to absorb ultraviolet (UV) rays [68] while utilizing them to degrade organic pollutants through photocatalyst reactions [69]. TiO_2_ compounds with particle sizes in the nano range have higher photocatalytic activity than the bulk phase. This can be attributed to the presence of a wider reactive surface area and an increase in the band gap energy through the mechanism of optical and electronic properties that depend on size; these properties are known as quantized particles (Q-particles) or quantum dots [70]. Photocatalytic activity can be analyzed using methylene blue in the wood samples. This substance is difficult to degrade, and its levels can be measured using a visible light spectrophotometer at a wavelength of 666 nm [71].

Titanium dioxide was studied extensively in the last two decades due to its strong activity in photocatalytic reactions. The photocatalytic activity of methylene blue substances can also be expressed as a percentage of their degradation [66]. The decrease in the concentration of methylene blue in the untreated mangium wood also occurred through adsorption, which can happen because the cellulose and lignin biomass in wood can behave as effective adsorbents. Adsorbents containing a high amount of cellulose can absorb basic dyes irreversibly [72]. The decrease in concentration of methylene blue in untreated sample also occurred because of heat from the UV radiation during the testing process [73].

#### 4.3.3. Evaluation of TiO_2_ Anatase Nanoparticle Efficiency on The Degradation of Methylene Blue

The calculation results of the E_EO_ (Figure 9) show that the availability of TiO_2_ nanoparticle can reduce the E_EO_ value. This is in line with the results of research by Ghavi et al. [51] that TiO_2_ can reduce the E_EO_ value because TiO_2_ nanoparticle can reduce the activation energy for the reaction process of pollutant degradation. This result indicated that the use of TiO_2_ nanoparticle and its application to mangium wood can reduce the cost of using electrical energy. The energy consumption of this research with TiO_2_ nanoparticle is 1166.68 kWh/m^3^, compared to the study of Varma et al. [74], who reported energy consumption with a value of 3000 kWh/m^3^. It turned out that our finding is smaller. This was attributed to the utilization of nano-sized TiO_2_, which had larger surface area and higher photocatalyst activity [75].

The decreased concentration of the methylene blue was linear in each treatment. This finding indicated that both on photocatalytic activity by anatase TiO_2_ nanoparticle and impregnation processes applied the pseudo-first order kinetic of dye removal kinetic. The treated mangium wood had higher gradient slopes than untreated (Figure 10). It proved that treated mangium wood had faster photocatalytic reaction compared with untreated at the same conditions. E_Eo_ and kinetic profile showed that anatase TiO_2_ nanoparticle is a reliable substance to degrade methylene blue as organic pollutant and catalyst to be used in photocatalytic reaction.

#### 4.3.4. UV-Vis Radiation Analysis

Protection from UV radiation is important for the performance of the wood substrate and the coating that protects it. Ultraviolet light causes degradation of wood substrates and coating polymers [76]. This result is in line with research conducted by Rassam et al. [24] indicating that the initiation reaction of wood degradation due to UV-Vis radiation and radiation heat transfer is a decrease in the intensity of the C-H aromatic and C=C functional groups in lignin compounds. This result proves that the synthesized anatase TiO_2_ nanoparticle can protect mangium wood from the degradation process due to UV-vis radiation even though it is still in the initiation reaction stage.

#### 4.3.5. FTIR

The functional groups were identified based on the C-H stretching from cellulose and hemicellulose, the aromatic C-H strain and C=C strain in the lignin framework, the O-H strain from cellulose, and the C-O strain from hemicellulose [77]. Based on the FTIR spectrum, mangium wood impregnated with 1% and 5% TiO_2_ nanoparticle (Figure 11) did not have the C-H functional group in the aromatic framework and had a decrease in the intensity of the C-H functional group. This may have happened because the TiO_2_ nanoparticle covered the surface of cellulose, lignin, and hemicellulose, which are polymers [78] that are the main structural building blocks of wood [79], and blocked infrared radiation. In a previous study, FTIR analysis of jabon wood impregnated with magnetite nanoparticle showed that nano magnetite caused a bulking effect when filling the wood pores; new functional groups were not identified, and new peaks from the infrared spectrum did not appear [80].

#### 4.3.6. XRD

Based on a comparative analysis of the standard JCPDS anatase TiO_2_ diffractogram card number 78-2486 [40], the peaks are anatase TiO_2_ compounds that successfully impregnated into mangium wood. In addition, XRD analysis was also used to identify the crystalline phase of wood after the impregnation with TiO_2_ nanoparticle. Based on the comparative analysis with the standard JCPDS cellulose diffractogram No. 03-0226 [81], these peaks were identified as cellulose, which still had the same crystal lattice as the untreated. This finding indicates that the impregnation treatment did not affect the crystalline structure of the cellulose; however, the degree of crystallinity was changed in comparison with the untreated wood, which was 71.98%.

This increase in crystallinity value was likely caused by the rearrangement of the cellulose structure in the amorphous phase owing to its interaction with the impregnant solution using water as a solvent and the effect of bulking inorganic particles along the cellulose surface [82]. Inorganic particles playing a role included the TiO_2_ nanoparticle. At higher TiO_2_ concentrations, the degree of wood crystallinity decreased owing to the amount of hydrophobic TiO_2_ [83] covering the surface of cellulose and consequently preventing rearrangement of cellulose in the amorphous phase. The results of the calculation of the crystal size of TiO_2_ using the Scherrer equation [44] were 61.64 nm for 1% TiO_2_ nanoparticle and 64.94 nm for 5% TiO_2_ nanoparticle. This value indicates that the particle size of TiO_2_ impregnated into the wood was smaller than the particle size prior to the impregnation process. This can have occurred because of the sonication treatment of the impregnated solution before the impregnation process, which can reduce the TiO_2_ particle size [84].

## 5. Conclusions

Anatase TiO_2_ nanoparticle has been successfully synthesized by hydrothermal method at low temperatures based on the results of characterization with spectrophotometer Uv-Vis, XRD, and FTIR. Anatase TiO_2_ nanoparticle has maximum wavelength of 362 nm and band gap energy of 3.40 eV.The values of E_Eo_ and kinetic profile proved that mangium wood impregnated with TiO_2_ nanoparticle has the ability to degrade methylene blue as the organic pollutant.Density and dimensional stability of treated mangium wood increased compared with untreated. Five percent TiO_2_ nanoparticle treatment obtained higher WPG, BE, ASE, and leachability values; however, lower WU values than that of untreated and 1% TiO_2_ nanoparticle.Based on XRD and FTIR analysis, TiO_2_ nanoparticle was successfully impregnated into mangium wood. Scanning electron microscopy–energy-dispersive X-ray spectroscopy analysis indicated that TiO_2_ nanoparticle covered the surface of the wood cells. The TiO_2_-impregnated mangium wood has a higher photocatalyst activity than untreated, indicating better protection from UV radiation and pollutants.

## Figures and Tables

**Figure 1 polymers-14-04463-f001:**
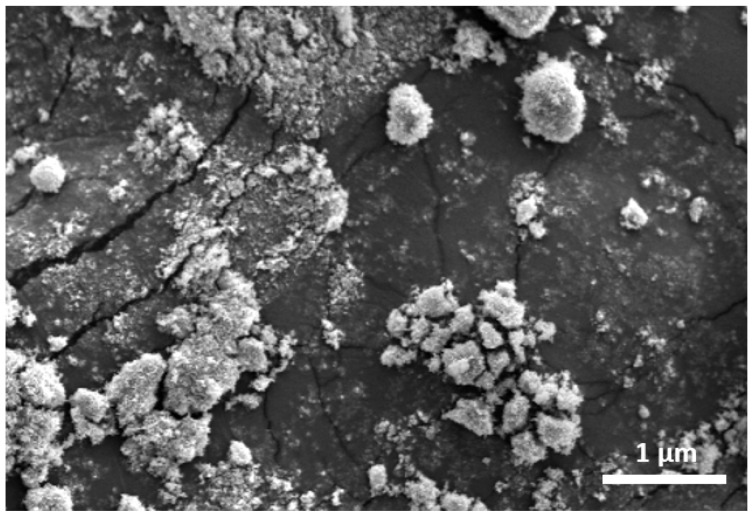
SEM images of TiO_2_ nanoparticle synthesized by the hydrothermal method.

**Figure 2 polymers-14-04463-f002:**
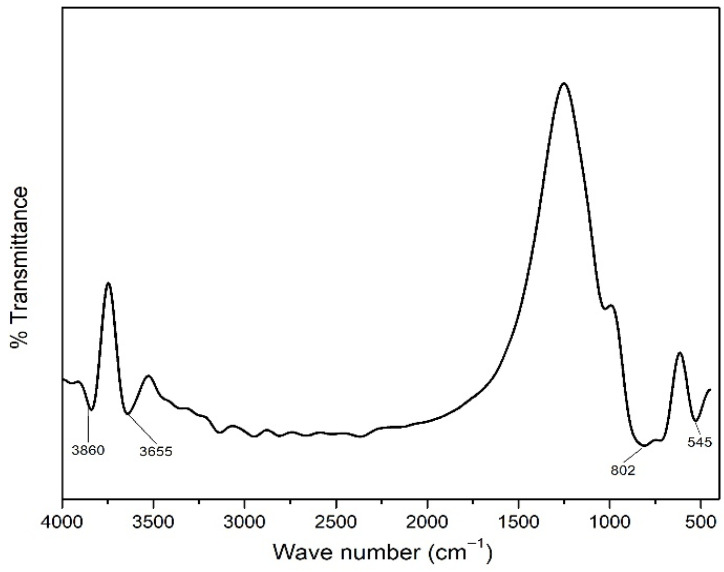
FTIR spectrum of TiO_2_ nanoparticle.

**Figure 3 polymers-14-04463-f003:**
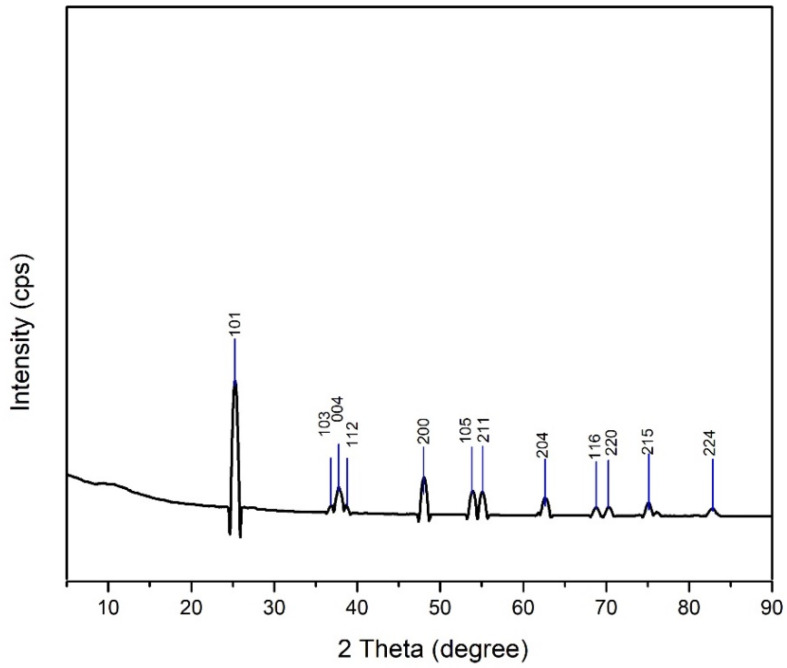
TiO_2_ nanoparticle diffractogram.

**Figure 4 polymers-14-04463-f004:**
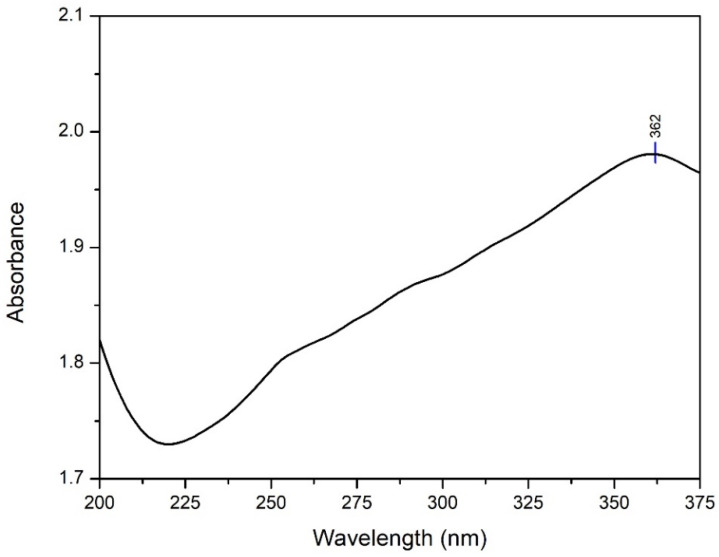
TiO_2_ nanoparticle absorption spectrum.

**Figure 5 polymers-14-04463-f005:**
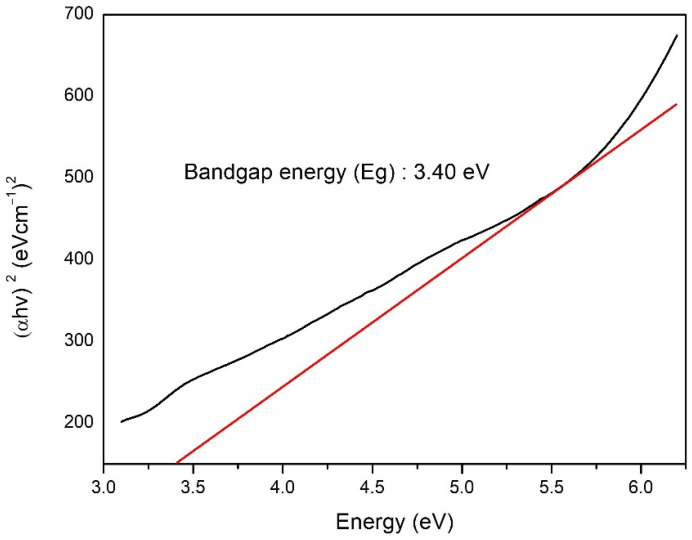
TiO_2_ nanoparticle band gap analysis curve with Tauc method.

**Figure 6 polymers-14-04463-f006:**
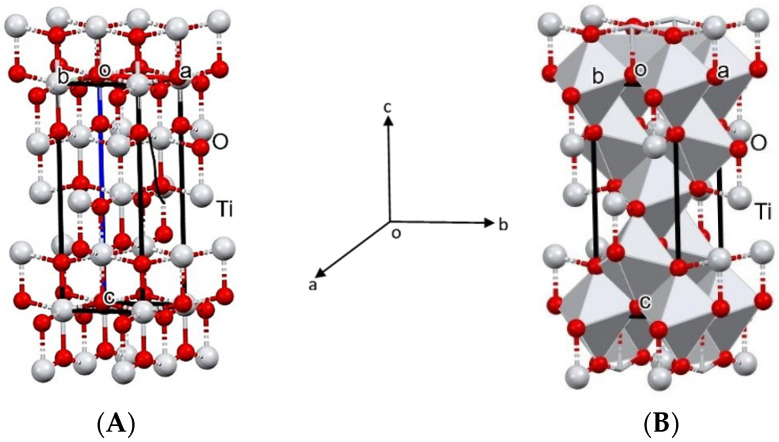
Crystal structure of anatases TiO_2_ nanoparticle in unit cells with (**A**) ball stick and (**B**) polyhedron patterns. a, b, c and o is the notation of the three-dimensional coordinate system contains an origin (o) and formed by three mutually perpendicular coordinate axes: the x-axis (a), y-axis (b), and the z-axis (c).

**Figure 7 polymers-14-04463-f007:**
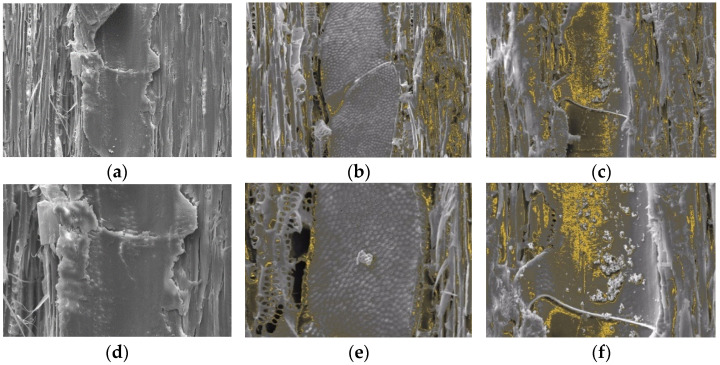
SEM images of untreated mangium wood 500× (**a**), mangium wood impregnated with 1% TiO_2_ nanoparticle 500× (**b**) and mangium wood impregnated with 5% TiO_2_ nanoparticle 500× (**c**), untreated mangium wood 1000× (**d**), mangium wood impregnated with 1% TiO_2_ nanoparticle 1000×, (**e**) and mangium wood impregnated with 5% TiO_2_ nanoparticle 1000× (**f**).

**Figure 8 polymers-14-04463-f008:**
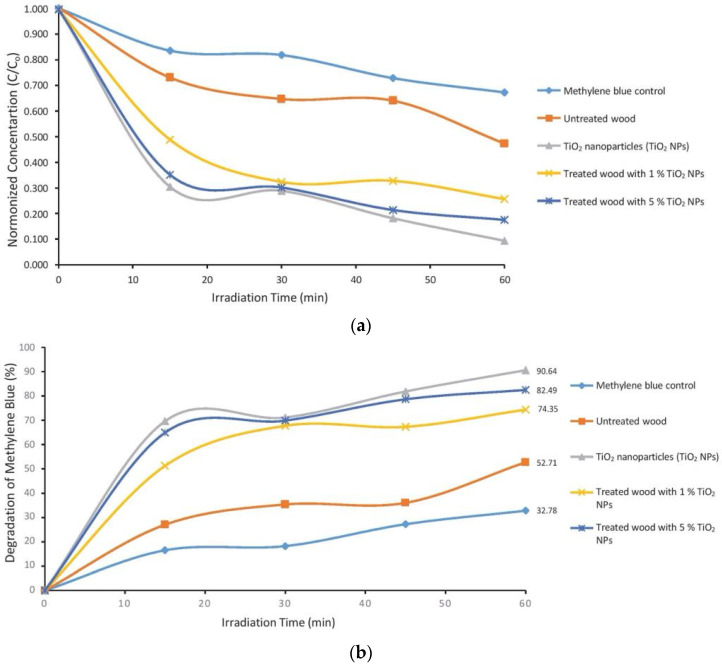
Curves (**a**) relative concentration of C/C_o_ and (**b**) percentage of degradation of methylene blue compounds.

**Figure 9 polymers-14-04463-f009:**
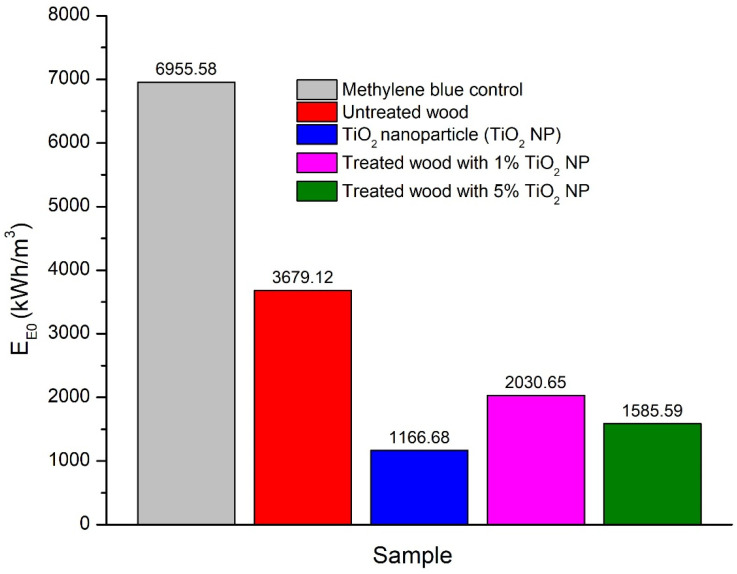
Energy consumption for the obtained samples.

**Figure 10 polymers-14-04463-f010:**
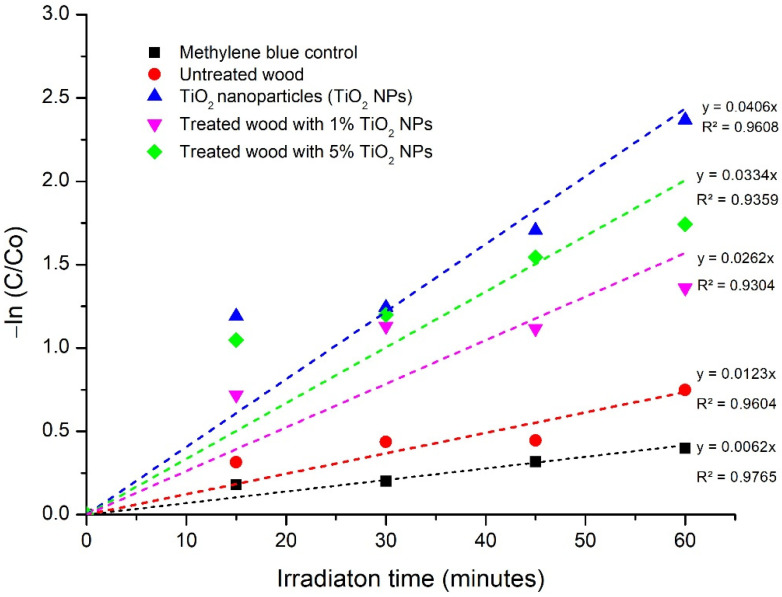
Reaction kinetics profile for the degradation of methylene blue in the presence of anatase TiO_2_ nanoparticle. Experimental conditions photocatalyst dose = 1 g/L; methylene blue = 10 ppm.

**Figure 11 polymers-14-04463-f011:**
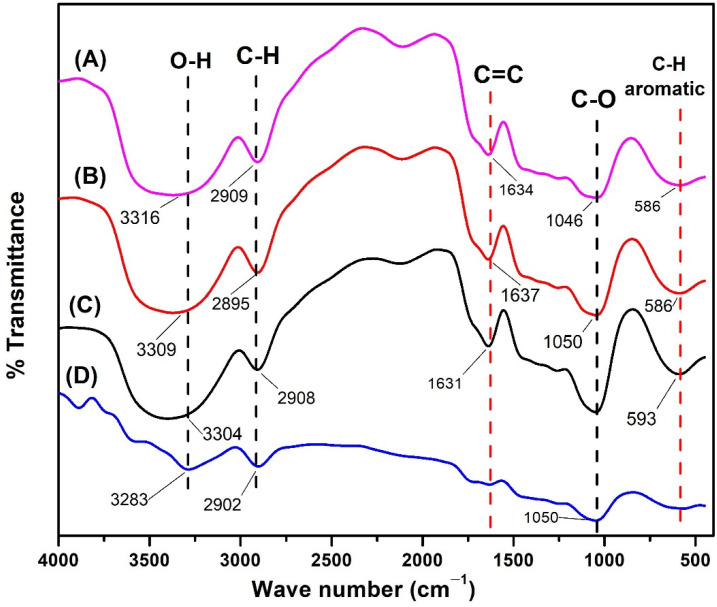
FTIR spectrum after UV-Vis radiation treatment on (**A**) impregnated mangium wood with 5% TiO_2_ nanoparticle, (**B**) impregnated mangium wood impregnated with 1% TiO_2_ nanoparticle, (**C**) untreated mangium wood without radiation, (**D**) untreated mangium wood with radiation.

**Figure 12 polymers-14-04463-f012:**
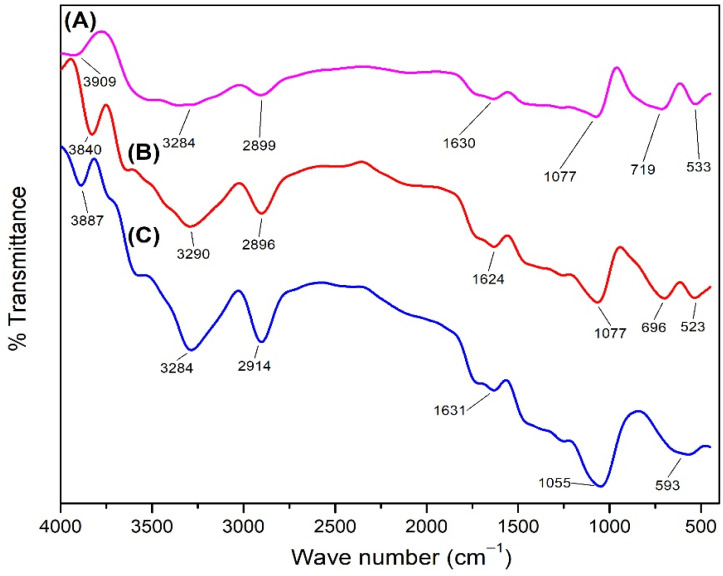
FTIR spectrum on (**A**) impregnated mangium wood with 5% TiO_2_ nanoparticle, (**B**) impregnated mangium wood with 1% TiO_2_ nanoparticle, (**C**) untreated mangium wood.

**Figure 13 polymers-14-04463-f013:**
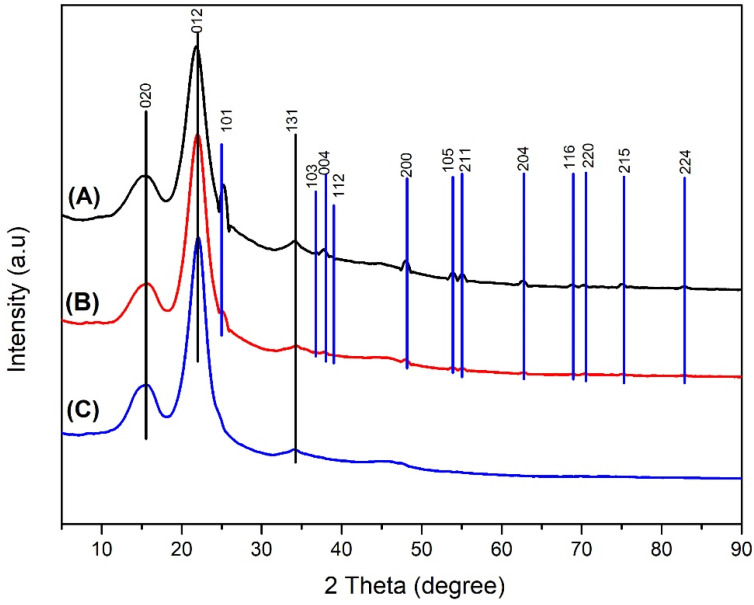
Diffractogram (**A**) impregnated mangium wood with 5% TiO_2_ nanoparticle (**B**) impregnated mangium wood with 1% TiO_2_ nanoparticle (**C**) untreated mangium wood.

**Table 1 polymers-14-04463-t001:** Value of dimensional stability and density of mangium wood.

Sample	Density	WPG	BE	WU	ASE	L
Untreated	0.54 ± 0.02 ^a^	0.21 ± 0.14 ^a^	1.04 ± 0.69 ^a^	50.29 ± 4.45 ^a^	0.00 ^a^	0.44 ± 0.19 ^a^
1% TiO_2_ nanoparticle	0.55 ± 0.02 ^ab^	0.53 ± 0.22 ^a^	1.14 ± 0.47 ^a^	49.08 ± 7.80 ^a^	15.46 ± 9.52 ^b^	0.57 ± 0.21 ^a^
5% TiO_2_ nanoparticle	0.57± 0.03 ^b^	2.76 ± 0.72 ^b^	2.10 ± 1.03 ^b^	46.66 ± 6.67 ^a^	21.76 ± 7.97 ^b^	0.43 ± 0.15 ^a^

^a,b^ Values followed by different letters are significantly different based on the Duncan test.

**Table 2 polymers-14-04463-t002:** SEM-EDX test results.

Sample	Ti (%Wt)
Untreated	-
1% TiO_2_ nanoparticle	0.35
5% TiO_2_ nanoparticle	4.69

**Table 3 polymers-14-04463-t003:** Data on crystallinity test results for mangium wood impregnated with TiO_2_ nanoparticle.

Sample	Degree of Crystallinity (%)
Untreated	71.98
1% TiO_2_ nanoparticle	74.17
5% TiO_2_ nanoparticle	65.13

## Data Availability

The data presented in this study are available on request from the corresponding author.

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
