# Peer review of "Physical Properties of Fast-Growing Wood-Polymer Nano Composite Synthesized through TiO2 Nanoparticle Impregnation"

_polymers, 2022, doi:10.3390/polym14204463_

Round 1
Reviewer 1 Report
In this study, nano TiO2 was produced by hydrothermal method. Authors analyzed the effect of nano TiO2 impregnation on density and dimensional stability of mangium. Using UV-Vis Spectrophotometer, FTIR and XRD to characterize the TiO2 nanoparticles. Impregnation synthesized nano-TiO2 has proven increased density, weight percent gain, bulking effect and anti-swelling efficiency. However, revision is necessary before publication.
My suggestion and comments are as follow:
1. TiO2 should be TiO2,please be consistent in expressing.
2. Some new references about TiO2 application should be added. Cell Reports Physical Science, 2022, 3, 101011; Journal of Catalysis 402 (2021) 289–299; Applied Catalysis B: Environmental, 2022, 315, 121550; Journal of Catalysis 401 (2021) 288–296
3. The world “synthezied” in concludes should be “synthesized”, please check the spelling.
4. Page 14, line 411. “The use of titanium oxide compounds can utilize UV radiation to degrade these pollutants.” How this conclusion was reached, please give relevant references to support it.
5. The abstract mentions that the 5% nano TiO2 treatment had the highest density values. The conclusion using only two concentrations as a comparison lacks some scientific validity.
6. Considering the practical application of the results, it is recommended to add energy consumption and economic projections.
7. The amount of data in this paper is relatively small, so please add some characterization and activity test
Author Response
COVER LETTER for REVIEWERS
In this study, nano TiO2 was produced by hydrothermal method. Authors analyzed the effect of nano TiO2 impregnation on density and dimensional stability of mangium. Using UV-Vis Spectrophotometer, FTIR and XRD to characterize the TiO2 nanoparticles. Impregnation synthesized nano-TiO2 has proven increased density, weight percent gain, bulking effect and anti-swelling efficiency. However, revision is necessary before publication.
Thank you very much for reviewers’ suggestions and corrections to our manuscript. We have made some changes and revisions according to reviewers’ suggestions. Our responds to each reviewer suggestions have been made also (represented by sentences in blue colour in this letter)
- TiO2 should be TiO2,please be consistent in expressing.
We have revised TiO2 by TiO2 (see revised manuscript)
- Some new references about TiO2 application should be added. Cell Reports Physical Science, 2022, 3, 101011; Journal of Catalysis 402 (2021) 289–299; Applied Catalysis B: Environmental, 2022, 315, 121550; Journal of Catalysis 401 (2021) 288–296
All references has been added to revised manuscript (see line 104, 105)
- The world “synthezied” in concludes should be “synthesized”,please check the spelling.
We have revised synthezied by synthesized (see revised manuscript)
- Page 14, line 411. “The use of titanium oxide compounds can utilize UV radiation to degrade these pollutants.”How this conclusion was reached, please give relevant references to support it.
We have added some information and the references regarding this (see line 548-552)
Pollutants consist of two types, namely organic and inorganic pollutants (Kermani et al., 2022). The use of titanium oxide compounds can protect biopolymer materials such as wood from UV radiation due to the strong ability to absorb ultraviolet (UV) rays (Guo et al., 2017) while utilizing them to degrade organic pollutants through photocatalyst reactions (Li et al., 2021)
- The abstract mentions that the 5% nano TiO2treatment had the highest density values. The conclusion using only two concentrations as a comparison lacks some scientific validity.
We have deleted the sentence from our abstract
- Considering the practical application of the results, it is recommended to add energy consumption and economic projections.
We have added energy consumption into our discussion (see line 369-404). Since our research is preliminary stage, we have not performed economic projection aspects in our study.
- The amount of data in this paper is relatively small, so please add some characterization and activity test
We have added result of energy consumption (EEo) and Kinetic profile into our discussion (see line 369-404).

Reviewer 2 Report
The work is good and contains a kind of modernity and could be accepted for publication, but after making a few modifications.
What is the difference between W1 in WPG (%) and W1 in WU (%)
The manuscript needs to improve the English language.
The method of preparing TiO2 must be described in detail
The conclusion should be rewritten so that it contains the important results and their importance to the researchers
Author Response
COVER LETTER for REVIEWERS
The work is good and contains a kind of modernity and could be accepted for publication, but after making a few modifications.
Thank you very much for reviewers’ suggestions and corrections to our manuscript. We have made some changes and revisions according to reviewers’ suggestions. Our responds to each reviewer suggestions have been made also (represented by sentences in blue colour in this letter)
What is the difference between W1 in WPG (%) and W1 in WU (%)
W1 in WPG is the same with Wi in WU. We have revised the sentences (see line 170-171)
The manuscript needs to improve the English language.
The manuscript has been read by our native speaker (Julia R. Barrett, MS, ELS - native English speaker and editor based in the United States
The method of preparing TiO2 must be described in detail
We have added more information regarding the method of preparing TiO2 nanoparticle (see line 123-132)
The conclusion should be rewritten so that it contains the important results and their importance to the researchers
We have revised the conclusions (see conclusion section on revised manuscript)

Round 2
Reviewer 1 Report
accepted
Reviewer 2 Report
After the authors completed all suggestion comments, the revised manuscript can be accepted.